# Psychotropic Drug Effects on Steroid Stress Hormone Release and Possible Mechanisms Involved

**DOI:** 10.3390/ijms23020908

**Published:** 2022-01-14

**Authors:** Zuzana Romanova, Natasa Hlavacova, Daniela Jezova

**Affiliations:** 1Institute of Experimental Endocrinology, Biomedical Research Center, Slovak Academy of Sciences, 84505 Bratislava, Slovakia; zuzana.romanova@savba.sk (Z.R.); natasa.hlavacova@savba.sk (N.H.); 2Department of Pharmacology and Toxicology, Faculty of Pharmacy, Comenius University Bratislava, 83232 Bratislava, Slovakia

**Keywords:** cortisol, aldosterone, antipsychotic drugs, lithium, molecular pathways, neurotransmitters

## Abstract

There is no doubt that chronic stress accompanied by adrenocortical stress hormone release affects the development and treatment outcome of several mental disorders. Less attention has been paid to the effects of psychotropic drugs on adrenocortical steroids, particularly in clinical studies. This review focuses on the knowledge related to the possible modulation of cortisol and aldosterone secretion under non-stress and stress conditions by antipsychotic drugs, which are being used in the treatment of several psychotic and affective disorders. The molecular mechanisms by which antipsychotic drugs may influence steroid stress hormones include the modulation of central and/or adrenocortical dopamine and serotonin receptors, modulation of inflammatory cytokines, influence on regulatory mechanisms in the central part of the hypothalamic-pituitary axis, inhibition of corticotropin-releasing hormone gene promoters, influencing glucocorticoid receptor-mediated gene transcription, indirect effects via prolactin release, alteration of signaling pathways of glucocorticoid and mineralocorticoid actions. Clinical studies performed in healthy subjects, patients with psychosis, and patients with bipolar disorder suggest that single and repeated antipsychotic treatments either reduce cortisol concentrations or do not affect its secretion. A single and potentially long-term treatment with dopamine receptor antagonists, including antipsychotics, has a stimulatory action on aldosterone release.

## 1. Introduction

Stress is unequivocally considered to be a significant contributing factor in the development and course of mental disorders. Stress is the body’s response to harmful events called stressors, stress stimuli, or stress situations, which aimed to help overcome demanding situations. Thus, the original message of the stress response is positive and useful. However, insufficiently coping with stress stimuli may lead to negative consequences. Important features during chronic stressor exposure are the unpredictability and inescapability of stress experience as well as individual differences in resilience and susceptibility [1].

Repeated or chronic exposure to stressors may induce a variety of positive and negative changes at the mental, neuroendocrine, and molecular levels [2]. Despite intensive research in the field of stress, an exact measurement of the stress load is still not possible. Nevertheless, there are a high number of stress markers, which can be analyzed by different methodological approaches in various tissues of humans and experimental animals (Figure 1). The development of new techniques allowed researchers to elucidate the molecular mechanisms that are involved and perform studies at the level of genes of individual stress system components and beyond.

The main steroid stress hormones, namely, glucocorticoids and mineralocorticoids, are produced and released into the circulation by the adrenal cortex. These stress hormones, represented by the mineralocorticoid aldosterone and the glucocorticoid cortisol (corticosterone in rodents), are crucial for many brain and somatic functions in health and disease. Both central and peripheral mechanisms participate in the regulation of cortisol and aldosterone secretion. Cortisol is the executive part of the hypothalamic–pituitary–adrenocortical (HPA) axis, which is under the control of higher brain centers, mediated by the release of hypothalamic corticotropic-releasing hormone (CRH). CRH reaches the anterior pituitary via hypothalamic portal circulation, triggers the release of adrenocorticotropic hormone (ACTH), and finally releases cortisol (corticosterone) from the adrenal glands [3]. Under stress conditions, ACTH also stimulates the aldosterone release. The secretion of aldosterone as the end component of the renin–angiotensin–aldosterone system is triggered by angiotensin II with physiological significance for the control of the water–electrolyte balance and blood pressure.

It has long been known that glucocorticoids induce a broad spectrum of action on metabolic, immune, cardiovascular and brain functions. The action of glucocorticoids is mediated via several, mainly established, molecular pathways. Evidence on glucocorticoid effects beyond the classical endocrine gland function has been accumulating and they are considered to play an important role in the pathophysiology of several mental disorders [4]. Clinical data on the possible involvement of the mineralocorticoid hormone aldosterone in the course and outcome of major depressive disorder have recently been obtained [5,6]. Aldosterone was considered not to influence any mental function due to the lack of enzyme 11beta-hydroxysteroid-dehydrogenase type 2 in the brain. However, certain brain regions contain mineralocorticoid receptors that preferentially bind aldosterone [5,7]. We have provided evidence on the causal relationship between hyperaldosteronism and anxiety, as well as depression-like behavior, in animal models [8,9].

While the association between stress hormones and mental disorders has been intensively studied, less attention has been given to the effects of the psychotropic drugs used in the treatment of mental disorders on stress hormone release, particularly in clinical studies. Psychotropic drugs are likely to influence regulatory mechanisms in the central part of the HPA axis as well as the signaling pathways of glucocorticoid and mineralocorticoid actions. This review focuses on knowledge related to the possible modulation of cortisol and aldosterone secretion under non-stress and stress conditions by antipsychotic drugs and lithium, which are used in the treatment of several psychotic and affective disorders.

## 2. Potential Molecular Mechanisms to Influence Adrenocortical Function

The mechanisms by which antipsychotics and other psychotropic drugs may affect adrenocortical functions have not been well-clarified. However, several possibilities could be taken into consideration (Figure 2).

### 2.1. Neurotransmitter Receptors

The main mechanism of antipsychotic drugs is the blockade of mesolimbic dopamine receptors. It was shown that dopamine, acting through both D1 and D2 receptors, exerts a stimulatory role on the activation of the HPA axis [10,11]. In preclinical studies, several D1 and D2 dopamine receptor agonists have been reported to increase plasma ACTH and corticosterone levels [12,13,14]. On the other hand, a reduction in dopaminergic activity, as a result of high antagonism at D2 receptors in the striatum following antipsychotic treatment, could directly lead to reduced ACTH and cortisol concentrations [15,16].

There is evidence that aldosterone secretion is subjected to a dopaminergic inhibitory mechanism directly at the level of the adrenal cortex. Pharmacological characterization and autoradiographic localization revealed D2-like (including D2, D3, and D4) receptors in the adrenal cortex, especially in the zona glomerulosa [17]. Administration of the D2 antagonist metoclopramide resulted in a rise in plasma aldosterone in humans [18,19] but failed to modify aldosterone concentrations in rats [20,21]. Wu et al. [22] demonstrated that activation of D4 receptors can increase aldosterone secretion, whereas an inhibitory effect is mediated via D2 receptors.

Several antipsychotics influence central serotoninergic transmission, particularly in the prefrontal cortex. Some, but not all, atypical antipsychotics are effective selective serotonin (5-HT) receptor subtypes, agonists or antagonists [23]. 5-HT2A/C receptors are known to play a role in the regulation of the HPA axis. The blockade of these receptors was shown to inhibit the secretion of glucocorticoids [24,25]. Preclinical studies with 5-HT agonists indicate that 5-HT2A receptors, rather than 5-HT2C receptors, affect the corticosterone concentration in rats [25,26]. Selected 5-HT receptors, particularly 5-HT4 and 5-HT7, are also present at the level of the adrenal cortex [27,28]. It is, however, uncertain whether these receptors are the target of antipsychotic drugs.

### 2.2. Cytokines

Another underlying mechanism of antipsychotic drug action on adrenocortical activity might be via pro-inflammatory cytokines. Pro-inflammatory cytokines can directly stimulate the HPA axis, causing an elevation in circulating glucocorticoid concentrations. At the same time, cytokines can inhibit glucocorticoid receptor function at multiple levels, including glucocorticoid receptor translocation and the induction of glucocorticoid receptor isoforms with a reduced capacity to bind ligands [29]. Some of these interactions, particularly glucocorticoid receptor binding to its DNA response element, occur in the nucleus. In the cytoplasm, interactions with several signaling proteins can occur, such as mitogen-activated protein kinase (MAPK), extracellular-signal-regulated kinase (ERK), c-Jun N-terminal kinase (JNK), Janus kinase (JAK), and IkB kinase beta [30]. All of the MAPKs have been identified as potential targets for the anti-inflammatory actions of glucocorticoids through blockage of their activating phosphorylations. Contrarily, cytokine-activated MAPK signaling can phosphorylate the glucocorticoid receptor itself, thereby modulating its turnover and its transcriptional activity, imposing an extra layer of glucocorticoid receptor function regulation [31]. Both preclinical and clinical studies have revealed that atypical antipsychotic drugs significantly suppressed the proinflammatory cytokines, and thereby reduced concentrations of plasma ACTH and glucocorticoids [32,33].

### 2.3. Endocrine Factors

A well-known undesired consequence of therapy with antipsychotic drugs acting via D2 receptors is the increased release of prolactin. This may represent another mechanism by which antipsychotics influence the HPA axis. This was suggested by the authors of a prenatal stress study in mice using prolonged treatment with the atypical antipsychotic drug paliperidone, which normalized the accumulation of lipid droplets in adrenocortical zona fasciculata (producing glucocorticoids) and altered plasma ACTH and corticosterone [34]. Importantly, prolactin may indirectly act on CRH neurons to downregulate the HPA axis [35].

Although the HPA axis is not a direct target of antipsychotic medication, both the prefrontal cortex and the hippocampus are involved in the action of antipsychotics and are linked to HPA axis functioning [36]. Atypical antipsychotics downregulate CRH expression [37], and this may be one of the mechanisms by which antipsychotics modulate HPA axis activity [38]. An in vitro study by Basta-Kaim et al. [38] showed that a majority of antipsychotic drugs markedly inhibited the basal activity of CRH gene promoter. The most potent effect in this respect was exerted by chlorpromazine, haloperidol, clozapine, and thioridazine, whereas promazine, risperidone, and raclopride were less active. The inhibition of CRH activity by clozapine and chlorpromazine mainly results from activation of the phosphoinositide-3-kinase–protein kinase B/Akt (PI3-K/Akt) pathway. However, the involvement of the Ca^2+^/calmodulin-dependent protein kinase (CaMK) and ERK-MAPK in the effects of some antipsychotic drugs on CRH gene activity should also be considered.

### 2.4. Trophic Factors and Adult Neurogenesis

One of the negative consequences of chronic exposure to stressors is a decrease in brain plasticity, demonstrated by a decreased expression of brain-derived neurotrophic factor (BDNF) and cell proliferation [39,40]. To date, there has been no clear evidence regarding the possible mediation of antipsychotic drug effects on adrenocortical steroids via brain neurotrophic factors. In a study in rats, treatment with olanzapine prevented prenatal stress-induced increase in corticosterone but not the stress-induced decrease in prefrontal cortex BDNF expression [41]. Osacka and colleagues reported that subchronic treatment with haloperidol and aripiprazole did not modify the effect of chronic unpredictable stressors on the BDNF and glucocorticoid receptor mRNA levels in the brain [42].

Antipsychotic drug treatments can help to overcome the negative consequences of chronic daily life stress situations by their effects at the level of the adult hippocampal neurogenesis, cell survival, and neuronal morphological complexity and, thus, indirectly influence steroid stress hormone release. The results of an elegant study in rats [43] using chronic unpredictable mild stress (seven weeks) and two antipsychotic drug treatments for three weeks are in support of the aforementioned suggestion. An enhancement of adult neurogenesis and neuronal survival was observed in rats treated with clozapine, whereas treatment with haloperidol promoted a downregulation of these processes. Furthermore, clozapine was able to re-establish the stress-induced impairments in neuronal structure and gene expression in the hippocampus and prefrontal cortex.

### 2.5. Hypothalamic Clock Gene Regulatory Mechanisms

Antipsychotic drugs that act to antagonize dopamine receptors are likely to affect the circadian system, as dopaminergic neurotransmission plays a role in the control of sleep and daily rhythms [44]. Antipsychotics may influence the quality of sleep and affect the circadian system [45]. Some studies propose that typical and atypical antipsychotics could have different effects on circadian rhythms [46,47,48,49,50,51,52]. A metanalysis by Moon et al. [45] showed that typical antipsychotics disrupt rest–activity rhythms, but atypical antipsychotics corrected the disrupted daily rhythmicity.

Biological rhythms are driven by a master pacemaker, the central Circadian Locomotor Output Cycles Kaput (CLOCK), located at the hypothalamic level in the suprachiasmatic nucleus (SCN), which dictates the time to peripheral satellite clocks. There is evidence from animal studies that haloperidol can directly influences the circadian clock. The acute administration of haloperidol to mice can induce clock gene *Per1* expression in the SCN via N-methyl-D-aspartic acid (NMDA) glutamate receptor subunits and cAMP-response element-binding protein (CREB) signaling [53]. Chronic treatment with haloperidol can suppress *Per1* expression across various brain regions [54] but has little or no effect on clock gene *Bmal1* expression. Lithium is also known to affect multiple aspects of the circadian rhythms. One well-established effect of lithium is lengthening of the circadian period, which seems to be mediated by glycogen synthase kinase-3 in the SCN [55]. At the level of the adrenal cortex, Chung et al. [56] demonstrated that adrenal-clock-dependent steroidogenesis and SCN-driven central mechanisms regulating the release of glucocorticoids cooperate to produce the daily circulatory rhythm of adrenocortical hormones.

### 2.6. Epigenetic and Glucocorticoid Receptor-Related Factors

According to the recent findings, epigenetic factors can also be involved in antipsychotic drug mechanisms of action with consequent modulations of stress hormone-induced regulatory effects. Brivio et al. [57] has reported that the antipsychotic drug lurasidone’s ability to normalize the expression of selected genes in stressed rats is associated with epigenetic modifications to their DNA methylation pattern. Lurasidone treatment prevented stress-induced hypermethylation of the glucocorticoid-responsive element of the glucocorticoid-receptor-responsive gene Gadd45*β* [57].

There is also evidence that some antipsychotics inhibit glucocorticoid-receptor-mediated gene transcription [58]. In vitro experiments showed that the inhibitory effect of chlorpromazine on the glucocorticoid-receptor-induced gene transcription depends on the inhibition of Ca(2+) influx and/or the inhibition of some calcium-dependent enzymes, e.g., phospholipase beta [59]. The inhibitory effect of clozapine on glucocorticoid receptor function may result from the inhibition of the phospholipase C/protein kinase C (PLC/PKC) pathway [58]. Indeed, chronic in vivo treatment with clozapine and some other antipsychotics (chlorpromazine or haloperidol) decreased PKC activity in discrete regions of the rat brain [60,61]. These data suggest that inhibition of the glucocorticoid-receptor-induced gene transcription by antipsychotics may be a mechanism by which these drugs mediate some effects on adrenocortical function.

### 2.7. Enzymes Involved in Steroidogenesis and Signaling Pathways

It is unclear whether antipsychotics can directly influence the steroidogenesis of adrenocortical stress hormones by influencing the enzymes involved in steroidogenesis. In a relatively old study [62], chlorpromazine was shown to inhibit cholesterol esterase activity in vitro, but the cholesterol esterase activity was not decreased in the adrenals of drug-treated rats.

Different molecular mechanisms can be expected to operate in the case of lithium, which is an inorganic ion. As a monovalent cation, lithium may mimic sodium actions in excitable tissues. Lithium inhibits several enzymes that are involved in signaling pathways. The best-known biochemical actions of lithium are the interference with inositol triphosphate formation and the inhibition of kinases. For example, lithium inhibits glycogen synthase kinase 3 (GSK3). GSK3 isoforms phosphorylate a number of key enzymes involved in pathways leading to apoptosis. These biochemical actions became textbook knowledge, together with the statement that the mechanism of therapeutic action of lithium in bipolar (manic-depressive) disorder is still not understood [63]. The biochemical changes induced by lithium may only indirectly influence the regulatory systems of adrenocortical stress hormone release, if at all. With respect to other hormones, lithium is known to inhibit the antidiuretic action of vasopressin [64] as well as the effect of the thyroid-stimulating hormone in the thyroid gland [65]. Treatment with lithium may affect activated HPA axis and corticosterone release via its effects on oxidative tissue damage and cytokine alterations, as shown in an animal model of mania using paradoxical sleep deprivation [66].

## 3. Antipsychotic Drugs, Lithium and Cortisol Release

### 3.1. Effects of Antipsychotic Drugs on Cortisol Release in Healthy Subjects

#### 3.1.1. Single Treatments

Measurements of circulating cortisol concentrations in response to single treatment with older antipsychotic drugs acting mainly by the blockade of dopamine D2 receptors did not provide consistent results (Table 1). A single oral treatment with haloperidol resulted in a prolonged reduction in salivary cortisol concentrations (Handley et al., 2016). The authors assumed that the reduction in salivary cortisol, as well as in plasma interleukin-6, was responsible for the observed increase in hippocampal cerebral blood flow. On the other hand, the same dose of haloperidol failed to modify plasma cortisol, measured repeatedly throughout the day [67]. Similarly, intravenous injections of amisulpride [68] and sulpiride [69,70,71] had no effect on cortisol secretion. Intravenous infusion of pimozide did not affect cortisol release during hypoglycemia stress [72]. On the contrary, oral pretreatment with haloperidol prevented a heat-stress-induced increase in salivary cortisol concentrations [73].

Consistent with the above-mentioned effect of haloperidol, a single oral administration of the atypical antipsychotic drug olanzapine resulted in lower serum/plasma cortisol concentrations [67,74]. The evaluation of possible mechanisms of olanzapine effects on metabolic markers allowed the authors to suggest that a single dose of olanzapine may invoke early changes in selected parameters of glucose and lipid metabolism.

The oral administration of two different doses of quetiapine, an atypical antipsychotic acting via different subtypes of serotonin, noradrenaline, and dopamine receptors, resulted in lower plasma cortisol concentrations compared to those after treatment with placebo [67,75]. The partial agonist of dopamine D2 receptors and selected serotonin receptor subtypes aripiprazole did not modify plasma cortisol levels under non-stress conditions [33]. Similarly, a single oral dose of zotepine in a small group of healthy subjects had no effect on plasma cortisol concentrations [76].

#### 3.1.2. Repeated Treatments

A limited number of trials in healthy volunteers have investigated repeated treatment with atypical antipsychotic drugs, with relatively consistent results (Table 1). A reduction in circulating cortisol was described following oral treatments with two distinct olanzapine formulations of the same dose for 8 days [77]. One group of authors focused on cortisol in stress repeatedly induced by exposure to acoustic stimuli during sleep (4–5 nights) compared to undisturbed sleep [78,79]. Oral treatment with both ziprasidone [78] and quetiapine [79] reduced urinary cortisol excretion in comparison with placebo.

### 3.2. Effects of Antipsychotic Drugs and Lithium on Cortisol Release in Patients

#### 3.2.1. Repeated Treatments

Obviously, as there are no studies using a prolonged antipsychotic treatment in healthy subjects, there is a low need to investigate the effects of single doses in patients. A very limited number of reports have been made on repeated, sub-chronic treatments (Table 2).

In a group of patients with major depressive disorder, one week of treatment with quetiapine positively influenced cortisol response to a dexamethasone/CRH test [80]. Cookson and colleagues reported an inhibitory action of a two-week treatment with pimozide on plasma cortisol in a very small group of patients with mania (*n* = 6) [81]. The same group of authors [82] observed a decrease in cortisol concentrations in a somewhat larger group of patients in the manic phase of bipolar disorder, treated with haloperidol.

#### 3.2.2. Long-Term Treatments

The dominant finding of studies evaluating serum cortisol in response to long-term treatment with antipsychotic drugs was an inhibitory effect on the HPA axis function (Table 2). Venkatasubramanian and colleagues compared serum cortisol values in the first episode of psychosis in patients treated with risperidone or olanzapine for three months and those of healthy volunteers and observed that antipsychotic therapy led to a decrease in cortisol concentrations [83]. Interestingly, the decrease in serum cortisol was accompanied by a rise in insulin-like growth factor-1 (IGF-1). Treatment with various antipsychotic drugs (risperidone, haloperidol, chlorpromazine, olanzapine) was used in patients newly diagnosed with schizophrenia for 8 weeks. Post-treatment serum cortisol concentrations were significantly lower than those at the baseline [84].

Treatment with risperidone was described to be more effective in the reduction in serum cortisol compared to haloperidol in patients with schizophrenia following therapy lasting for 12 weeks [85]. Reduced cortisol concentrations were also observed in patients with schizophrenia treated with olanzapine [86,87]. However, there are papers reporting the lack of effect of long-term antipsychotic medication on plasma cortisol using clozapine and haloperidol [88] or olanzapine [89].

A lowering effect on cortisol concentrations was also observed in patients on a long-term (12 months) treatment with lithium. This relatively old study [90] was performed on 53 patients; however, they were suffering from miscellaneous psychiatric conditions (unipolar affective disorder, bipolar disorder, recurrent depression, cycloid psychosis, schizoaffective disorder). The authors observed a decrease in morning cortisol concentrations, thus disturbing the circadian rhythmicity of cortisol secretion. Another study investigated patients with bipolar affective disorders treated with lithium alone for six months or more [91]. The study was oriented toward metabolic functions and used a five-hour oral glucose tolerance test. At the end of the test, they observed significantly increased cortisol concentrations in lithium-treated patients compared to untreated patients, apparently as a result of the hypoglycemia stress induced by lithium. Bschor and colleagues [92,93] evaluated the effects of four-week lithium augmentation or monotherapy on cortisol and ACTH response to the dexamethasone/CRH test in patients with unipolar depression. Their main observation was significantly higher cortisol and ACTH response to the dexamethasone/CRH test following lithium therapy compared to the baseline, indicating the stimulatory action of lithium on the HPA axis. The mechanisms of this lithium effect are unknown [92,93].

With respect to neuroendocrine stress response, few studies evaluated cortisol release following acute stressors. The patients with bipolar disorder treated with various antipsychotics (*n* = 49) exhibited a blunted salivary cortisol response to the Trier social stress test for groups compared to healthy subjects (*n* = 48) [94]. A recent study has evaluated salivary cortisol response to a similar stress test. It is, however, difficult to judge the influence of antipsychotic drug medications, as the treatments were not described in sufficient detail [95]. Antipsychotic medication for one year did not influence ability to cope with daily life stressors [96].

## 4. Antipsychotic Drugs, Lithium and Aldosterone Release

Published information on the effect of antipsychotics on aldosterone is very limited (Table 3). Surprisingly, several relatively old studies investigated aldosterone response to a single medication with an antipsychotic drug in small groups of healthy subjects or patients with schizophrenia. Thus, plasma aldosterone concentrations significantly increased in healthy women in the follicular phase of the menstrual cycle following the administration of sulpiride [97]. It is important to note that aldosterone concentrations fluctuate throughout the menstrual cycle and are associated with mood disturbances [98]. Most studies published to date do not take this fact into consideration. Similarly, enhanced plasma aldosterone concentrations were induced by the intravenous administration of chlorpromazine in male patients with schizophrenia [99]. A single intravenous infusion of haloperidol was reported to induce an increase in plasma aldosterone in treatment-naïve psychotic patients requiring acute care [100] or no change in a small group of healthy and schizophrenic subjects [101]. A single dose of lithium carbonate to healthy male and female subjects failed to modify plasma aldosterone concentrations [102].

Long-term treatment with antipsychotic drugs (mainly atypical antipsychotics) is likely to contribute to enhanced serum aldosterone concentrations compared to those in drug-naïve patients during the first episode of psychosis [103]. In a study aiming to investigate imbalanced water homeostasis in patients with schizophrenia, volume-regulatory hormones were measured during abdominal surgery. The concentrations of plasma aldosterone in response to surgery were significantly lower in chronically antipsychotic-treated patients compared to subjects in the control group [104].

There are only few studies evaluating changes in aldosterone concentrations in response to treatment with lithium carbonate, with inconsistent results. A small group of patients were subjected to lithium therapy for three months and their plasma concentrations of aldosterone were compared with those of patients treated with lithium for several years [105]. Both longitudinal (three months) and long-term transversal treatment with lithium failed to change aldosterone concentrations, which were within normal limits. On the other hand, other authors reported elevated plasma aldosterone concentrations in patients with affective disorders taking lithium prophylactically (above six years) compared to those in healthy subjects [106]. Interestingly, high concentrations of plasma aldosterone were found in patients suffering from lithium intoxication [107].

## 5. Conclusions

It is evident that antipsychotic drugs either reduce cortisol concentration or do not affect its secretion. According to the review article by Borges and colleagues [108], most of the studies in patients with schizophrenia suggest that atypical antipsychotics reduce cortisol concentrations to a greater extent than typical antipsychotics. In our opinion, there is not enough evidence in the current literature to support this suggestion.

A single treatment with dopamine receptor antagonists, such as antipsychotic drugs, has a stimulatory action on aldosterone release. To date, insufficient evidence is available to confirm this stimulatory influence on aldosterone secretion by long-term treatment with antipsychotic drugs.

## Figures and Tables

**Figure 1 ijms-23-00908-f001:**
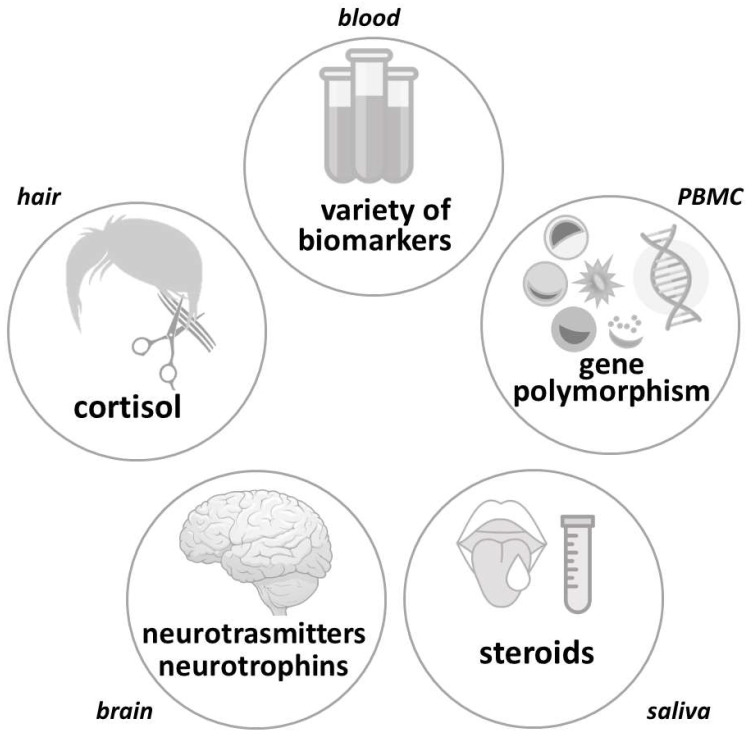
Different biomarkers related to stress, measured in select tissues and biological fluids in humans and experimental animals. PBMC—peripheral blood mononuclear cells.

**Figure 2 ijms-23-00908-f002:**
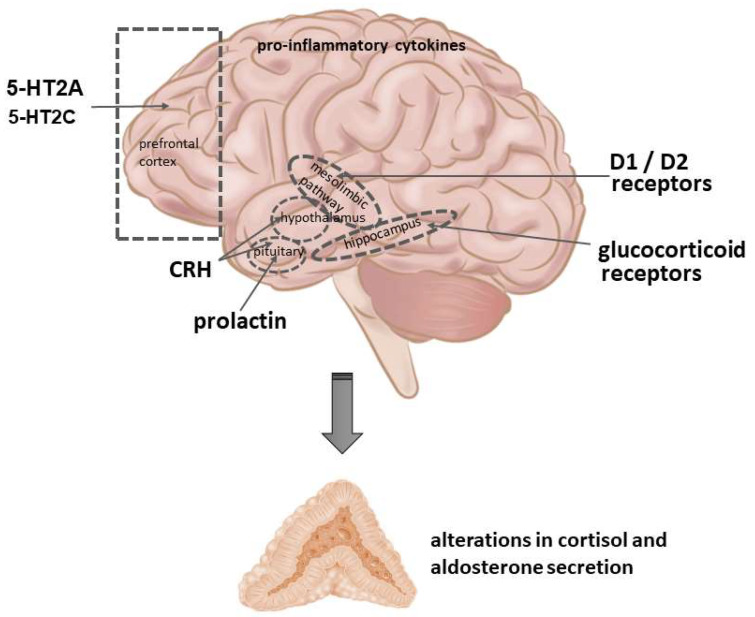
Mechanisms by which antipsychotic drugs may affect cortisol (corticosterone in rodents) and aldosterone secretion. CRH—corticotropin-releasing hormone; D1, D2—dopamine receptor subtypes, 5-HT2A, 5-HT2C—serotonin receptor subtypes.

**Table 1 ijms-23-00908-t001:** Summary of findings from studies investigating changes in cortisol concentrations following antipsychotic drug treatment in healthy subjects.

Study	Drug (s)	Duration of Treatment	Participants	Biological Fluid	Cortisol Secretion	Refs.
Handley et al. (2016)	haloperidol (3 mg)	single dose	healthy (*n* = 17)	saliva	↓	[33]
	aripiprazole (10 mg)	single dose	healthy (*n* = 17)	saliva	–	
Cohrs et al. (2006)	haloperidol (3 mg)	single dose	healthy (*n* = 11)	plasma	–	[67]
	quetiapine (50 mg)	single dose	healthy (*n* = 11)	plasma	↓	
	olanzapine (5 mg)	single dose	healthy (*n* = 11)	plasma	↓	
Wetzel et al. (1994)	amisulpride (20 mg)	single dose	healthy (*n* = 8)	plasma	–	[68]
	amisulpride (100 mg)	single dose	healthy (*n* = 8)	plasma	–	
Laakmann et al. (1984)	sulpiride (100 mg)	single dose	healthy (*n* = 6)	plasma	–	[69]
von Bahr et al. (1991)	sulpiride (200 mg)	single dose	healthy (*n* = 12)	serum	–	[70]
	remoxipride (100 mg)	single dose	healthy (*n* = 12)	serum	–	
de Konnig and de Vries (1995)	sulpiride (400 mg)	single dose	healthy (*n* = 19)	plasma	–	[71]
	haloperidol (3 mg)	single dose	healthy (*n* = 19)	plasma	–	
Henning et al. (1995)	haloperidol (3 mg)	single dose	healthy (*n* = 40)	saliva	↓ stress-induced	[73]
Hahn et al. (2013)	olanzapine (10 mg)	single dose	healthy (*n* = 12)	serum	↓	[74]
de Borja Goncalves Guerra et al. (2005)	quetiapine (150 mg)	single dose	healthy (*n* = 15)	plasma	↓	[75]
von Bardeleben et al. (1987)	zotepine (25 mg)	single dose	healthy (*n* = 6)	plasma	–	[76]
Jezova-Repcekova et al. (1979)	pimozide (4 mg)	two doses	healthy (*n* = 8)	plasma	– stress-induced	[72]
Vidarsdottir et al. (2009)	olanzapine (10 mg)	8 days	healthy (*n* = 12)	plasma	↓	[77]
Meier et al. (2005)	ziprasidone (40 mg)	5 days	healthy (*n* = 11)	urine	↓ stress-induced	[78]
Cohrs et al. (2004)	quetiapine (25 mg)	4 days	healthy (*n* = 13)	urine	↓ stress-induced	[79]
	quetiapine (100 mg)	4 days	healthy (*n* = 13)	urine	↓ stress-induced	

Abbreviations: ↓ = decrease; – = no change.

**Table 2 ijms-23-00908-t002:** Summary of findings from studies investigating changes in cortisol concentrations following antipsychotic drug treatment in patients.

Study	Drug (s)	Duration of Treatment	Participants	Biological Fluid	Cortisol Secretion	Refs.
Sarubin et al. (2014)	quetiapine (300 mg)	1 week	MDD patients (*n* = 23)	blood	↓	[80]
Cookson et al. (1980)	pimozide (32 mg)	2 weeks	BD patients (*n* = 6)	plasma	↓	[81]
Cookson et al. (1985)	haloperidol (2.5–40 mg)	3 days	BD patients (*n* = 31)	plasma	↓	[82]
Venkatasubramanian et al. (2010)	risperidone (4–10 mg)	12 weeks	SZ patients (*n* = 17)	serum	↓	[83]
	olanzapine (7.5–15 mg)	12 weeks	SZ patients (*n* = 14)	serum	↓	
Woldesenbet et al. (2021)	risperidone, haloperidol, chlorpromazine, olanzapine, modecate	8 weeks	SZ patients (*n* = 34)	serum	↓	[84]
Zhang et al. (2005)	risperidone (6 mg)	12 weeks	SZ patients (*n* = 41)	serum	↓	[85]
	haloperidol (20 mg)	12 weeks	SZ patients (*n* = 37)	serum	↓	
Mann et al. (2006)	olanzapine (15–20 mg)	4 weeks	SZ patients (*n* = 10)	serum	↓	[86]
Tanaka et al. (2008)	olanzapine (20 mg)	16 weeks	SZ patients (*n* = 28)	plasma	↓	[87]
Breier et al. (1994)	clozapine (400 mg)	10 weeks	SZ patients (*n* = 11)	plasma	–	[88]
	haloperidol (20 mg)	10 weeks	SZ patients (*n* = 15)	plasma	–	
Jakovljevic et al. (2007)	olanzapine (5–20 mg)	22 weeks	SZ patients (*n* = 12)	plasma	–	[89]
Smigan and Perris (1984)	lithium (unspecified)	1 year	various patients (*n* = 53)	serum	↓	[90]
Shah et al. (1986)	lithium (unspecified)	6 months	BD patients (*n* = 9)	serum	↑ stress-induced	[91]
Bschor et al. (2002)	lithium (900 mg)	4 weeks	MDD patients (*n* = 24)	plasma	↑ stress-induced	[92]
Bschor et al. (2011)	lithium (900 mg)	4 weeks	MDD patients (*n* = 30)	plasma	↑ stress-induced	[93]
Houtepen et al. (2015)	various antipsychotics	unspecified	BD patients (*n* = 49)	saliva	↓ stress-induced	[94]
Rossini Gajsak et al. (2021)	various antipsychotics	unspecified	SZ patients (*n* = 53)	saliva	↓ stress-induced	[95]
Vaessen et al. (2018)	various antipsychotics	1 year	psychotic patients (*n* = 49)	saliva	– stress-induced	[96]

Abbreviations: ↓ = decrease; – = no change; ↑ = increase; MDD = Major Depressive Disorder; BD = Bipolar Disorder; SZ = Schizophrenia.

**Table 3 ijms-23-00908-t003:** Summary of findings from studies investigating changes in aldosterone concentrations following antipsychotic drug treatment in healthy subjects and patients.

Study	Drug (s)	Duration of Treatment	Participants	Biological Fluid	Aldosterone Secretion	Refs.
Costa et al. (1980)	sulpiride (100 mg)	single dose	healthy (*n* = 7)	plasma	↑	[97]
Robertson and Michelakis (1975)	chlorpromazine (10 mg)	single dose	SZ patients (*n* = 12)	plasma	↑	[99]
Liberini et al. (1996)	haloperidol (2 mg)	single dose	SZ patients (*n* = 7)	plasma	↑	[100]
Warner et al. (1992)	haloperidol (0.02 mg/kg BW)	single dose	SZ patients (*n* = 12)	serum	–	[101]
Shirley et al. (1991)	lithium (300 mg)	single dose	healthy (*n* = 15)	plasma	–	[102]
	lithium (600 mg)	single dose	healthy (*n* = 15)	plasma	–	
Ustohal et al. (2018)	various antipsychotics	unspecified	SZ patients (*n* = 36)	serum	↑	[103]
Kudoh et al. (1998)	chlorpromazine, perphenazine	long-term	SZ patients (*n* = 18)	plasma	↓ stress-induced	[104]
Pedersen et al. (1977)	lithium (600 mg)	3 months	BD patients (*n* = 8)	plasma	–	[105]
	lithium (unspecified)	3 months–20 year	BD patients (*n* = 27)	plasma	–	
Stewart et al. (1988)	lithium (unspecified)	0.5–18 years	BD patients (*n* = 16)	plasma	↑	[106]

Abbreviations: ↓ = decrease; – = no change; ↑ = increase; BD = Bipolar Disorder; SZ = Schizophrenia; BW = Body Weight.

## Data Availability

Not applicable.

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
