# Peer review of "Psychotropic Drug Effects on Steroid Stress Hormone Release and Possible Mechanisms Involved"

_ijms, 2022, doi:10.3390/ijms23020908_

Round 1

Reviewer 1 Report

Romanova and Coworkers  focused  this review on the effects of  psychotropic drugs on  stress hormone release and possible mechanisms involved, paying particular attention to the papers appeared in the literature on this topic and following the lines of their previous  studies in animal and human  on the role of HPA hormones  in several mental disorders and on the effects of psycotropic drugs.. Even if the authors affirm that" less attention has been given to the effects of psychotropic drugs on adrenocortical steroids",  searching for  "psychotropic drugs and hypothalamic-pituitary- adrenal axis" on PubMed, 16 pages with 157 papers are referenced. I think this review adds few news on this subject, and also the role of mineralcorticoid hormones, in particular aldosterone, has been sufficiently elucidated in previous papers. Moreover, discussing the potential molecular mechanisms to influence adrenocortical function, the authors shouldn't ignore the role of the main hypothalamic clock gene which regulates from its site in the suprachiasmatic nucleous(SCN), the rhythmic secretion of HPA axis secretions in cooperation with peripheral clock genes . Even because  the stressors may alter the chronoorganization of this axis  impairing the rhythmic release of stress hormone , and some psychotropic drugs,  which act also at hypothalamic levels where is located the SCN, could impair or restore the circadian mechanisms involved in the rhythmic regulation  of  this axis.

Minor concerns:

- The last sentence of the Abstract is inconclusive.

 - Some important previous papers on this argument are missing in bibliography.

Author Response

Changes made in the revised version of the manuscript ID: ijms-1526406 entitled "Psychotropic drug effects on steroid stress hormone release and possible mechanisms involved" by Romanova et al. 

The changes made in the revised version of the manuscript are highlighted using the track changes mode and by red color in this letter. 

Reviewer 1

Point 1

Even if the authors affirm that" less attention has been given to the effects of psychotropic drugs on adrenocortical steroids", searching for "psychotropic drugs and hypothalamic-pituitary-adrenal axis" on PubMed, 16 pages with 157 papers are referenced.

 Response

The reviewer is right, but we had in mind the clinical studies. The animal studies are important for searching the mechanisms involved and they are the basis of the subchapter on molecular mechanisms in this review.

The following change has been made in the Abstract:

…. Less attention has been given to the effects of psychotropic drugs on adrenocortical steroids, particularly in clinical studies.

The following change has been made in the last paragraph of the Introduction:

…..intensively studied, less attention, particularly in clinical studies, has been given to the effects of the psychotropic drugs used in the treatment…….

Point 2

I think this review adds few news on this subject, and also the role of mineralcorticoid hormones, in particular aldosterone, has been sufficiently elucidated in previous papers. Moreover, discussing the potential molecular mechanisms to influence adrenocortical function, the authors shouldn't ignore the role of the main hypothalamic clock gene which regulates from its site in the suprachiasmatic nucleous (SCN), the rhythmic secretion of HPA axis secretions in cooperation with peripheral clock genes. Even because the stressors may alter the chronoorganization of this axis impairing the rhythmic release of stress hormone, and some psychotropic drugs, which act also at hypothalamic levels where is located the SCN, could impair or restore the circadian mechanisms involved in the rhythmic regulation of this axis.

 Response

We highly appreciate these very constructive suggestions. We agree with the reviewer that this important topic was missing in the original version of the review. Motivated by the reviewer suggestion, we have added a new subchapter to the “Potential molecular mechanisms to influence adrenocortical function”

The following text has been added:

Hypothalamic clock gene regulatory mechanisms

Antipsychotic drugs which act to antagonize dopamine receptors are likely to affect the circadian system as dopaminergic neurotransmission plays a role in the control of sleep and daily rhythms (Ashton and Jagannath, 2020). There are indices that antipsychotics influence the quality of sleep, and affect the circadian system (Moon et al. 2021). Some studies propose that typical and atypical antipsychotics could have different effects on circadian rhythms (Ayalon et al., 2002, Wirz-Justice et al. 1997, Wirz-Justice, Haug, 2001, Hwang et al., 2017, Matsui et al., 2017, Suzuki et al., 2018, Wirz-Justice et al., 2000). Metanalysis by Moon et al. (2021) showed that typical antipsychotics disrupt rest-activity rhythms, but atypical antipsychotics corrected disrupted daily rhythmicity.

Biological rhythms are driven by a master circadian clock, the suprachiasmatic nucleus (SCN), located in the hypothalamus. There is evidence from animal studies that haloperidol can directly influences the circadian clock. Acute administration of haloperidol to mice can induce clock gene Per1 expression in the SCN via N-methyl-D-aspartic acid (NMDA) glutamate receptor subunits and cAMP-response element-binding protein (CREB) signaling (Viyoch et al., 2005). Chronic treatment with haloperidol can suppress Per1 expression across various brain regions (Coogan et al., 2011), but has no or small effect on clock gene Bmal1 expression. Lithium is also known to affect multiple aspects of circadian rhythms. One well-established effect of lithium is lengthening the circadian period, which seems to be mediated by glycogen synthase kinase-3 in the SCN (Vadnie and McClung, 2017). At the level of the adrenal cortex, Chung et al. (2017) demonstrated that adrenal clock-dependent steroidogenesis and SCN-driven central mechanism regulating the release of glucocorticoids cooperate to produce the daily circulatory rhythm of adrenocortical hormones.

The following references have been added to the reference list:

Ashton, A.; Jagannath, A. Disrupted Sleep and Circadian Rhythms in Schizophrenia and Their Interaction With Dopamine Signaling. Front Neurosci 2020, 14, 636.

Ayalon, L.; Hermesh, H.; Dagan, Y. Case Study of Circadian Rhythm Sleep Disorder Following Haloperidol Treatment: Reversal by Risperidone and Melatonin. Chronobiol Int 2002, 19, 947–959.

Coogan, A.N.; Papachatzaki, M.M.; Clemens, C.; Baird, A.; Donev, R.M.; Joosten, J.; Zachariou, V.; Thome, J. Haloperidol Alters Circadian Clock Gene Product Expression in the Mouse Brain. World J Biol Psychiatry 2011, 12, 638–644.

Hwang, J.Y.; Choi, J.-W.; Kang, S.-G.; Hwang, S.H.; Kim, S.J.; Lee, Y.J. Comparison of the Effects of Quetiapine XR and Lithium Monotherapy on Actigraphy-Measured Circadian Parameters in Patients With Bipolar II Depression. J Clin Psychopharmacol 2017, 37, 351–354.

Chung, S.; Lee, E.J.; Cha, H.K.; Kim, J.; Kim, D.; Son, G.H.; Kim, K. Cooperative Roles of the Suprachiasmatic Nucleus Central Clock and the Adrenal Clock in Controlling Circadian Glucocorticoid Rhythm. Sci Rep 2017, 7, 46404.

Matsui, K.; Takaesu, Y.; Inoue, T.; Inada, K.; Nishimura, K. Effect of Aripiprazole on Non-24-Hour Sleep-Wake Rhythm Disorder Comorbid with Major Depressive Disorder: A Case Report. Neuropsychiatr Dis Treat 2017, 13, 1367–1371.

Moon, E.; Lavin, P.; Storch, K.-F.; Linnaranta, O. Effects of Antipsychotics on Circadian Rhythms in Humans: A Systematic Review and Meta-Analysis. Prog Neuropsychopharmacol Biol Psychiatry 2021, 108, 110162.

Suzuki, H.; Hibino, H.; Inoue, Y.; Mikami, K. Effect of Aripiprazole Monotherapy in a Patient Presenting with Delayed Sleep Phase Syndrome Associated with Depressive Symptoms. Psychiatry Clin Neurosci 2018, 72, 375–376.

Vadnie, C.A.; McClung, C.A. Circadian Rhythm Disturbances in Mood Disorders: Insights into the Role of the Suprachiasmatic Nucleus. Neural Plast 2017, 2017, 1504507.

Viyoch, J.; Matsunaga, N.; Yoshida, M.; To, H.; Higuchi, S.; Ohdo, S. Effect of Haloperidol on MPer1 Gene Expression in Mouse Suprachiasmatic Nuclei. J Biol Chem 2005, 280, 6309–6315.

Wirz-Justice, A.; Cajochen, C.; Nussbaum, P. A Schizophrenic Patient with an Arrhythmic Circadian Rest-Activity Cycle. Psychiatry Res 1997, 73, 83–90.

Wirz-Justice, A.; Haug, H.J.; Cajochen, C. Disturbed Circadian Rest-Activity Cycles in Schizophrenia Patients: An Effect of Drugs? Schizophr Bull 2001, 27, 497–502.

Wirz-Justice, A.; Werth, E.; Savaskan, E.; Knoblauch, V.; Gasio, P.F.; Müller-Spahn, F. Haloperidol Disrupts, Clozapine Reinstates the Circadian Rest-Activity Cycle in a Patient with Early-Onset Alzheimer Disease. Alzheimer Dis Assoc Disord 2000, 14, 212–215.

Point 3

The last sentence of the Abstract is inconclusive.

 Response

We agree with the reviewer and the last sentence of the Abstract has been deleted from the revised version.

The last sentence of the Abstract now reads as follows:

A single and potentially also long-term treatment with dopamine receptor antagonists, including antipsychotics, has a stimulatory action on aldosterone release.

 Point 4

Some important previous papers on this argument are missing in bibliography.

 Response

In addition to the new references mentioned above, we have added to the bibliography the following references:

de Koning, P.; de Vries, M.H. A Comparison of the Neuro-Endocrinological and Temperature Effects of DU 29894, Flesinoxan, Sulpiride and Haloperidol in Normal Volunteers. Br J Clin Pharmacol 1995, 39, 7–14.

Mann, K.; Rossbach, W.; Müller, M.J.; Müller-Siecheneder, F.; Pott, T.; Linde, I.; Dittmann, R.W.; Hiemke, C. Nocturnal Hormone Profiles in Patients with Schizophrenia Treated with Olanzapine. Psychoneuroendocrinology 2006, 31, 256–264.

Tanaka, K.; Morinobu, S.; Ichimura, M.; Asakawa, A.; Inui, A.; Hosoda, H.; Kangawa, K.; Yamawaki, S. Decreased Levels of Ghrelin, Cortisol, and Fasting Blood Sugar, but Not n-Octanoylated Ghrelin, in Japanese Schizophrenic Inpatients Treated with Olanzapine. Prog Neuropsychopharmacol Biol Psychiatry 2008, 32, 1527–1532.

von Bahr, C.; Wiesel, F.A.; Movin, G.; Eneroth, P.; Jansson, P.; Nilsson, L.; Ogenstad, S. Neuroendocrine Responses to Single Oral Doses of Remoxipride and Sulpiride in Healthy Female and Male Volunteers. Psychopharmacology (Berl) 1991, 103, 443–448.

The relevant information from these papers has been inserted into the tables and the following sentence has been added to the text:

……therapy lasting 12 weeks (Zhang et al. 2005). Reduced cortisol concentrations were observed also in patients with schizophrenia treated with olanzapine (Mann et al. 2006, Tanaka et al. 2008). However, there are papers reporting a lack of effect of long-term…..

Reviewer 2

Comments to the Author

,,Psychotropic drug effects on steroid stress hormone release and possible mechanisms involved written'' by Zuzana Romanova, Natasa Hlavacova , Daniela Jezova is well desribed paper.

 Response

We thank the reviewer for his/her positive judgments.

 Point 1

I have following minor remarks Figure 1 details instead of blood: variety of markers name biomarkers associated with HPA axis eg. cortisol, ACTH, CRH

Response

We agree with the reviewer that there is a variety of biomarkers, which could be mentioned. Figure 1, however, represents a schematic drawing of the existence of various biological fluids and selected tissues, and mentioning many biomarkers would make it blind.

Point 2

brain name also cortisol, CRH, while in brain is placed question mark

Response

Thank you, the question mark has been deleted from Figure 1.

Point 3

line 70 remove the word only.

 Response

Thank you, the word “only” has been removed.

Point 4

Please, cite the reference that there is lack of lack of the enzyme 11beta-hydroxysteroid-dehy-71 drogenase type 2 in the brain.

 Response

Thanks to this reviewer comment we realized that not only the citation but also a statement was missing in the original version of the manuscript.

The following statement has been added to the text:

…., aldosterone was considered not to influence any mental function due to the lack of the enzyme 11beta-hydroxysteroid-dehydrogenase type 2 in the brain. However, certain brain regions contain mineralocorticoid receptors that bind preferentially aldosterone (Geerling et al. 2006; Jezova et al. 2019). We have provided

The missing reference has been added to the reference list:

Geerling, J.C.; Kawata, M.; Loewy, A.D. Aldosterone-Sensitive Neurons in the Rat Central Nervous System. J Comp Neurol 2006, 494, 515–527.

Point 5

The arrow from D1, D2 receptors should come to the mesolimbic pathway The arrow from CRH should come also to pituitary.

Response

Figure 2 has been corrected according to the reviewer’s suggestions.

 Point 6

In 21 reference should be mRNA instead MRNA.

Response

With thank the reviewer for noting this mistake. It has been corrected.

 Point 7

On line 370, there is a parenthesis closure and no opening.

Response

With thank the reviewer for noting this mistake. It has been corrected.

We thank the reviewers for their valuable comments.  

Daniela Jezova

Reviewer 2 Report

,,Psychotropic drug effects on steroid stress hormone release and possible mechanisms involved written'' by Zuzana Romanova , Natasa Hlavacova , Daniela Jezova is well desribed paper. I have following minor remarks Figure 1 details instead of blood: variety of markers name biomarkers associated with HPA axis eg. cortisol, ACTH, CRH brain name also cortisol, CRH, while in brain is placed question mark line 70 remove the word only Please, cite the reference that there is lack of lack of the enzyme 11beta-hydroxysteroid-dehy-71 drogenase type 2 in the brain The arrow from D1, D2 receptors should come to the mesolimbic pathway The arrow from CRH should come also to pituitary In 21 reference should be mRNA instead MRNA On line 370, there is a parenthesis closure and no opening

Author Response

(The authors gave the same response as above.)

Round 2

Reviewer 1 Report

I think that the manuscript has been deeply revised and improved , in particular by adding the paragraph"Hypothalamic clock gene regulatory mechanisms".

Concerning this, I suggest to modify  the sentence at line 205  " Biological rhthms are driven by a master circadian clock, the suprachiasmatic nucledous(SCN), located in the hypothalamus" to:  biological rhythms are driven by  a master pace-maker, the central CLOCK((Circadian Locomotor Output Cycles Kaput), located at hypothalamic level in the suprachiasmatic nucleous(SCN),, which dictates the time to peripheral satellite clocks.

Author Response

Changes made in the re-revised version of the manuscript ID: ijms-1526406 entitled "Psychotropic drug effects on steroid stress hormone release and possible mechanisms involved" by Romanova et al. 

 The changes made in the re-revised version of the manuscript are highlighted using the track changes mode and by red color in this letter. 

 REVIEWER 1

Point 1

I think that the manuscript has been deeply revised and improved , in particular by adding the paragraph"Hypothalamic clock gene regulatory mechanisms".

 Response

We thank the reviewer for the positive judgment.

Point 2

Concerning this, I suggest to modify  the sentence at line 205  " Biological rhthms are driven by a master circadian clock, the suprachiasmatic nucledous(SCN), located in the hypothalamus" to:  biological rhythms are driven by  a master pace-maker, the central CLOCK((Circadian Locomotor Output Cycles Kaput), located at hypothalamic level in the suprachiasmatic nucleous(SCN),, which dictates the time to peripheral satellite clocks.

 Response

Thanks for the suggestion, the sentence has been modified accordingly:

Biological rhythms are driven by a master pace-maker, the central CLOCK (Circadian Locomotor Output Cycles Kaput), located at hypothalamic level in the suprachiasmatic nucleus (SCN), which dictates the time to peripheral satellite clocks. 

Sincerely,

Daniela Jezova